# cfDNA Changes in Maximal Exercises as a Sport Adaptation Predictor

**DOI:** 10.3390/genes12081238

**Published:** 2021-08-12

**Authors:** Kinga Humińska-Lisowska, Jan Mieszkowski, Andrzej Kochanowicz, Błażej Stankiewicz, Bartłomiej Niespodziński, Paulina Brzezińska, Krzysztof Ficek, Eglė Kemerytė-Ivanauskienė, Paweł Cięszczyk

**Affiliations:** 1Faculty of Physical Education, Gdańsk University of Physical Education and Sport, 80-336 Gdansk, Poland; andrzej.kochanowicz@awf.gda.pl (A.K.); paulina.brzezinska@awf.gda.pl (P.B.); pawel.cieszczyk@awf.gda.pl (P.C.); 2Faculty of Physical Education and Sport, Charles University, 162 52 Prague, Czech Republic; 3Institute of Physical Education, Kazimierz Wielki University, 85-064 Bydgoszcz, Poland; blasta@ukw.edu.pl (B.S.); barnie@ukw.edu.pl (B.N.); 4Faculty of Physiotherapy, The Jerzy Kukuczka Academy of Physical Education in Katowice, 40-065 Katowice, Poland; krzysztof.ficek@galen.pl; 5Academy of Education, Vytautas Magnus University, 03111 Vilnius, Lithuania; egle.kemeryte-ivanauskiene@vdu.lt

**Keywords:** cfDNA, aerobic exercise, exercise load, anaerobic exercise

## Abstract

Changes of circulating free plasma DNA (cfDNA) are associated with different types of tissue injury, including those induced by intensive aerobic and anaerobic exercises. Observed changes are dependent from induced inflammation, and thus it may be a potential marker for athletic overtraining. We aimed to identify the response of cfDNA to different types of exercise, with association to exercise intensity as a potential marker of exercise load. Fifty volunteers (25 athletes and 25 physically active men) were assigned to the study and performed maximal aerobic (Bruce test) and anaerobic (Wingate Anaerobic Test) test. Blood samples for cfDNA analysis were collected at four time-points: before, 2–5 min after, 30 min after and 60 min after each type of maximal physical activity. The two-way ANOVA revealed a significant effect of group factor on serum cfDNA concentrations (32.15% higher concentration of cfDNA in the athletes). In turn the results of the post hoc test for the interaction of the repeated measures factor and the group showed that while the concentration of cfDNA decreased by 40.10% in the period from 30 min to 60 min after exercise in the control group, the concentration of cfDNA in the group of athletes remained at a similar level. Our analysis presents different responses depending on the intensity and duration of exercise. Our observations imply that formation of cfDNA is associated with response to physical activity but only during maximal effort.

## 1. Introduction

High intensity exercises lead to transient muscle fibre damage, performance deterioration, increased inflammation processes [1] and oxidative stress induced by the generation of free radicals [2]. All those factors are associated with muscle injury, and occur during normal competition, especially when the duration or intensity of sport competition is very high. In a situation in which the accumulation of exercise load is very high, and decreased athletic performance is induced, the presence of muscle injury and increased inflammation processes are also observed. It is called ‘overtraining’ and may lead to increased immune system activity with elevated secretion of inflammatory factors [3,4,5,6].

There is rising evidence suggesting that, in addition to its health benefits, intensive exercise may have potential adverse effects on the immune system [3,7]. Therefore, clarification on this matter is necessary for training recommendations or personalized training load adjustment. Preventing injuries during training is critical to optimum performance. While exercise has both acute and chronic effects on the immune system, most biomarkers are insufficient and show only modest association with susceptibility to tissue injury. Adaptation mechanisms against soft tissue injuries as well as accelerating recovery time after different types of exercise are still poorly understood and these biomarkers still need to be established. In recent years cell-free DNA (cfDNA) has emerged as a potential biomarker of injury level, inflammation, cellular stress and cell death, generating more and more interest across various biomedical disciplines. Its changes are correlated with the presence of tissue injury, immune system activation, and even cancer [8,9,10,11,12]

Such pathological conditions have already been reported in patients with sepsis, myocardial infarction, cardiovascular diseases, autoimmune disorders, cancer, or metabolic disorders, but also in patients after trauma [13].

Circulating cfDNA is defined as double-stranded DNA molecules found in body liquids such as plasma, serum or urine, resulting from biological fragmentation upon stress induction. cfDNA circulating in the body has several fragment sizes. Most cfDNA molecules display short fragments (~150–180 bp), which are mainly derived from apoptosis via the activation of cellular endonucleases leading to the cleavage of chromatin DNA into internucleosomal fragments [14]. Long cfDNA fragments of >10 kbp are thought to be generated by necrosis. Another source of cfDNA is neutrophil extracellular traps (NETs), which play an active role in the innate immune system response to both pathogens and physical exercise [15]. Physiologically, low concentrations of cfDNA (2.5–27.0 ng/mL) circulate in the plasma/serum from healthy individuals before being cleansed by the liver [16]. Elevated concentration of circulating cfDNA in the bloodstream is considered to be a novel molecular biomarker, a hallmark manifestation of inflammatory response (not only associated with diseases, but also physical exhaustion)—whether acute or chronic. This has prompted sports medical researchers to investigate the role of cDNA as a potential marker of exercise induced-metabolic changes in a healthy human body after—i.e., resistance training, marathon run, continuous treadmill running, incremental exercise, rowing, soccer, or strength training [17,18,19,20]. It is already known that high levels of physical activity are associated with metabolic and mechanical muscular damage, leukocyte inflammatory response and DNA damage caused by oxidative stress, which in turn may induce an increase in cfDNA concentration [10]. Elevated concentrations of cfDNA have already been observed immediately after acute exercise and exhibiting a rapid return to baseline levels, whereas typical biomarkers of skeletal muscle injury (C-reactive protein, uric acid, creatine kinase) display delayed kinetics compared with the cfDNA peak response [10].

Additionally, exercise parameters such as the duration and intensity of aerobic running have already been positively correlated with selected markers of muscle damage [21], which indicates the enormous potential of cfDNA as a biomarker for exercise load in the aerobic and the anaerobic state.

Strong data can be found in the literature on the effect of exercise on changes in cfDNA, but little attention has been paid to the study of internal and extracellular mechanisms resulting from adaptive changes in the body caused by many years of training, as well as the type of exercise [9,10,13,18]. Therefore, the aim of the research was to assess the changes of circulating cfDNA in association with intensive aerobic and anaerobic exercises. These changes may be associated with exercise intensity and may indicate the physical effort exerted during the type of exercise. In conclusion, measurement of cfDNA may be a good predictor of specific activity preparation; therefore, measurement of its changes may indicate the precise stage preparation and may be a useful tool in athlete preparation diagnosis.

## 2. Materials and Methods

### 2.1. Experimental Overview

Participants were assigned to two groups: the athlete group and the control group. The study protocol involved maximal aerobic and anaerobic testing and cfDNA isolation during and after testing. During the initial visit, data on the subject’s age, body composition, and height were collected. Measurements began with maximal anaerobic testing, followed 14 days later by a maximal aerobic test. All volunteers were examined by a professional physician. A sample of venous blood was obtained at four time-points: before, up to 5, 30, and 60 min after each type of maximal physical activity. All laboratory analyses were performed at the Gdansk University of Physical Education (Gdansk, Poland).

### 2.2. Participants

Twenty-five male athletes (football players) (aged 20.41 ± 1.36) and an equal number of physically active men—control group (aged 21.78 ± 1.98) were recruited for the study. The controls’ and athletes’ height and body mass were 180.12 ± 6.46 cm and 178.60 ± 6.1 cm and 79.40 ± 13.34 kg and 76.03 ± 6.78 kg, respectively. Participant characteristics are presented in Table 1. Participants were recruited through letter of intent from AZS AWFiS Sport academic club and among the population of students from the city of Gdańsk, Poland. The football players were a part of the same team competing in the Polish third league (Europe) in the 2018/2019 season. Of those players, 4 were external defenders, four were central defenders, 10 were midfielders, 5 were wingers, and 2 were central forwards. Players were not involved in any other training programs aside from the training regimen imposed by the coach. The cohort of physical active men (control group) were participating in physical activity during studies in the course Physical Education and Sport. Students attended only courses such as basketball, gymnastics, athletics, or swimming one time per week (90 min per course).

During the whole experiment, all participants were instructed to maintain their everyday diet, and were asked to refrain from vigorous exercise and avoid caffeine and alcohol consumption during the 48 h preceding the testing date. Food was not consumed during testing and water was available ad libitum. None of the participants had any history of known diseases or reported intake of medication due to illnesses in the six months before the experiment.

The study protocol was accepted by the Bioethics Committee for Clinical Research of the Regional Medical Society in Gdansk (KB-27/18) and conducted according to the Declaration of Helsinki. Written informed consent was obtained from all study participants, who were also informed about the possibility of withdrawing consent at any time and for any reason. Prior to participation, subjects were informed about the study procedures.

### 2.3. Body Composition

Body composition was assessed with an InBody 770 multi-frequency bioelectrical impedance (BIA) device (InBody 770, Cerritos, CA, USA). Participants were instructed to come to the laboratory in the early morning, after at least a 10-h fast and no prior exercise either the day before or the day of the test. All testing was performed between 7:00 and 8:00. During the analysis, subjects stood on the metal platform of the device barefoot with the soles of their feet on the surface of the electrodes. Subjects then grasped the handles of the unit with their thumb and fingers to maintain direct contact with the electrodes. They stood still for ~1 min while maintaining their elbows fully extended and their shoulder joint abducted to approximately a 30-degree angle. All relevant data are shown in Table 1.

### 2.4. Measurement of Fitness Level

#### 2.4.1. Anaerobic Fitness Level—Wingate Anaerobic Test

To measure maximal anaerobic effort (MAnE) double repeated Wingate Anaerobic test (WAnT) was conducted on a cycle ergometer (Monark 894E, PeakBike, Sweden). For each participant, the saddle height was adjusted so the knee remained slightly flexed after the completion of the downward stroke (with final knee angle approximately 170–175). Toe clips were used to ensure that the participants’ feet were held firmly in place and in contact with the pedals. Before any experimental testing, each individual completed a standardized warm-up on the cycloergometer (five min at 60 rpm, 1 W/kg). Each participant was required to pedal with maximum effort for a period of 30 s against a fixed resistive load of 75 g/kg of total body mass as recommended by Bar-Or [22]. After that, participants had a 30 s break and WAnT was repeated in the same manner.

#### 2.4.2. Aerobic Fitness Level—Bruce Treadmill Test

To measure maximal aerobic effort (MAE), Bruce Protocol was performed by the participants. Testing started from an initial warm-up period (at low workload), followed by progressive graded exercise with increasing loads, an adequate time interval in each level, and a post-maximum effort recovery period (again at low workload). The protocol was composed of six stages, initially at 2.72 km at an inclination percentage of 10% degree. From that moment, velocity was increased but not according to the same increment ratio, remaining between 0.8 kph and 1.28 kph [23]. Although velocity did not present uniform increase during the test, the inclination which started at a 10% grade had its values altered always at a 2% degree ratio. Time was constant for each stage, with velocity and inclination alterations introduced every three minutes. The interruption criterion was associated with fatigue, difficulties in breathing, muscular tiredness, chest pain, or any factor limiting the effort. Moreover, the participants were under the Quark CPET Cardio Pulmonary Exercise Testing Unit, using a face mask with a gas and flow sensor ‘turbine flow meter’.

### 2.5. cfDNA Plasma Sampling, Isolation, and Measurement

4.9 mL of whole blood was collected in EDTA-collection tubes (S-Monovette K3 EDTA, Sarstedt, Germany) at four time-points: before the warm-up, up to 5 min after, 30 min after, and 60 min after each type of maximal physical effort. Time-points were chosen in accordance with Breithach et al. [10] and Haller et al. [24]. Blood samples were centrifuged (1600× *g*, 10 min.) within 2 h after collection (until centrifugation stored at 4 °C). The obtained plasma samples were transferred to fresh tubes and subjected to the second centrifugation (16,000× *g*, 10 min.) to remove the cell debris. Clear plasma samples were again transferred to fresh tubes and aliquots were stored at −80 °C until further analysis. NucleoSpin cfDNA XS kit (Macherey-Nagel, Duren, Germany) was used to extract cfDNA from 200 µL of plasma samples according to the rapid manufacturer’s protocol. cfDNA elution was performed with 30 µL of elution buffer and followed by 5 min incubation. Upon isolation, cfDNA quantification was performed using a method described in Jylhava et al. [25]. Briefly, the level of cfDNA in plasma was measured from the blood sample using the Qubit fluorometer 4.0 (Thermo Fisher Scientific, Waltham, MA, USA) in combination with the Qubit 1X dsDNA HS Assay Kit (Thermo Fisher Scientific, Waltham, MA, USA) according to the manufacturer’s instructions. For all cfDNA extractions, 2 µL of sample was diluted in 198 ul Qubit 1X dsDNA HS working solution before measurement.

### 2.6. Statistical Analysis

Descriptive statistics included mean ± standard deviation (SD) for all measured variables. In addition, 95% confidence intervals (CI) were calculated. Student′s t-test was used to investigate intergroup differences in physical and blood count characteristics as well as characteristics of maximal anaerobic (MAnE) and aerobic effort (MAE). Two-way ANOVA with repeated measures (RM: baseline, up to 5 min after, 30 and 60 min after) was used to investigate the cfDNA level after MAnE and MAE in the participants depending on their fitness level (group: control, athletes). Another set of two-way ANOVA with repeated measures was performed to investigate the differences in cfDNA changes from baseline to immediately after, as well as 30 and 60 min after, two maximal efforts (effort: MAnE, MAE) depending on fitness level of participants (group: control, athletes). Tukey’s post hoc test was performed to assess differences in particular subgroups. In addition, the effect size was estimated by eta-squared statistics (ƞ^2^). Values equal to or more than 0.01, 0.06, and 0.14, indicated a small, moderate, and large effect, respectively. Shapiro–Wilk and Levene’s test were performed to check the normal distribution and the homogeneity of variance, respectively. All analyses were performed with Statistica 12 (StatSoft Inc., Tulsa, OK, USA) and the level of significance was set at *p* = 0.05.

## 3. Results

The blood count characteristics are presented in Appendix A. The results of the Student’s *t*-test showed a significantly lower level of MPV (4.58%) and higher level of HDL (19.12%) in the athlete group compared to the control group. There were no significant differences in the rest of the examined blood count parameters (Appendix A). All men in both groups successfully completed two different maximal physical tests. The Student’s t-test analysis showed the significant differences in MAnE between groups (Appendix A). The athlete group had a 7.86% and 6.11% higher relative mean and peak power in comparison to the control group during the first WAnT, respectively. In the second WAnT, after 30 s of rest, a similar decrease in the relative mean and maximum power was noted in both groups. During MAE, the athlete group had a 15.24% and 9.07% higher maximum oxygen uptake and a maximum ventilation compared to the control group, respectively (Appendix A).

Serum levels of cfDNA before and after MAnE and MAE are presented in Figure 1. When analysing the time up to 5 min after MAnE, 88% of the athletes were responders, while in the control group there were only 17% (overall: 56%). In the case of MAE, 64% of athletes and 56% of the control group were responders (overall: 62%) to the applied effort in terms of cfDNA increase. The two-way ANOVA revealed a significant effect of group and RM factors in MAnE on serum cfDNA concentrations (Appendix A). For the main group, the factor showed a 32.15% higher concentration of cfDNA in the athlete group compared to the control group. In turn, a significant influence of the RM factor was shown by the greatest increase (38.19%) in cfDNA concentration up to 5 min after exercise compared to the baseline values and 60 min after collection (61.16%). The analysis of variance also showed a significant interaction of the group and RM factor in MAnE. The results of the post hoc test showed a 102.62% increase in the concentration of cfDNA up to 5 min after MAnE compared to the baseline values in the athlete group, while in the control group there was a gradual decrease, achieving about 49.79% lower concentrations of cfDNA 60 min after MAnE in compared to the baseline. As a result of the above changes, 68.47% higher cfDNA concentrations were also noticed up to 5 min after MAnE in the athlete group compared to the control group.

Analysis of the variance of cfDNA concentrations before and after the MAE showed a significant influence of the RM factor and its interaction with the group factor (Appendix A). The RM post hoc test results showed a significant increase in cfDNA concentration up to 5 min after (27.84%) and 30 min after exercise (27.77%) compared to baseline values. In turn the results of the post hoc test for the interaction of the RM factor and the group showed that while the concentration of cfDNA decreased by 40.10% in the period from 30 min to 60 min after exercise in the control group, the concentration of cfDNA in the group of athletes remained at a similar level (Figure 1).

The changes in cfDNA from the baseline visit to up to 5 min after and 30 min and 60 min after maximum exercise are shown in Figure 2. Analysis of variance showed a significant effect of the group factor on changes in cfDNA concentration up to 5 min after and 60 min after maximum exercise (Appendix A). Regardless of the type of exercise, the difference between the groups was 49% for the change in cfDNA concentration up to 5 min after exercise. A two-fold difference between the groups was noted for the change in cfDNA concentration 60 min after exercise (104.34%). The significant interaction of the group factor and the type of effort was noted only for the change in cfDNA concentration up to 5 min after exercise (Figure 2A). The results of the post hoc test showed a 94.03% smaller increase in the control group in cfDNA concentration immediately after MAnE compared to the athlete group.

## 4. Discussion

The main goal of the current study was to define whether cfDNA changes were associated with maximal aerobic and anaerobic effort, and whether those changes indicate specific sport adaptation and/or physical activity preparation for different types of activity.

Both analysed groups successfully completed two different maximal physical tests—double repeated WanT and Bruce protocol. Test analysis showed that the athlete group had a 7.86% and 6.11% higher relative mean and peak power in comparison to control group during the first WAnT. Similar conclusions were observed in the work of researchers like Rotstein et al. [26], Reaburn et al. [27], and many others. In the second WAnT, after 30 s of rest, a similar decrease in the relative mean and maximum power was noted in both groups. This is probably due to the fact that the energy expenditure spent on the implementation of the first task in both populations was analogous. During MAE, the athlete group had a 15.24% higher maximum oxygen uptake and a 9.07% higher maximum ventilation compared to the control group. Such conclusions are summed up in the meta-analysis [28], where training experience is directly correlated with maximal oxygen intake and higher results, which are mostly observed in training individuals.

There are many studies that show that inflammation processes are associated with physical activity, especially when it is done in maximal effort conditions [29,30,31]. Most analyses are based on serum exerkine level, because it is a well-established and well understood activity [29]. In most populations adapted to physical activity, we observe the specific mechanism of body adjustment associated to the type of training that the person does over the course of a lifetime. Some populations will show aerobic, others anaerobic adaptations. Those adaptations will correspond to time and characteristics of training, and take place both at the cellular and molecular level [32].

However, analysing specific mechanisms of adaptations, we came to the conclusion that there is still a lack of knowledge that could show the relationship between adaptation to training and the specific nature of the post-exercise response. According to our knowledge this is the first study which presents a comparison involving two different maximal physical effort tests, between an athlete and a control group in search of a new molecular biomarker for exercise load in aerobic and anaerobic conditions. Our analyses focused on measurement of plasma level of cfDNA. It seems that cfDNA could be an interesting alternative for the assessment of the level of inflammation and specific post-training adaptations in a population of physically active people. Its concentration in the bloodstream manifests inflammatory response (not only associated with diseases, but also physical exhaustion) plasma cfDNA levels subsequently increase during vigorous physical activity, such as endurance and resistance exercise [10].

Haller at al. [24] revealed an increase of cfDNA level during aerobic running below the lactate steady state, which depended on the training intensity and duration. They revealed that cfDNA concentrations peaked immediately after acute exercise and about 1 h post-exercise returned to baseline levels. That is in accordance with our results among athletes after MAnE. The greatest increase (38.19%) in cfDNA concentration was observed up to 5 min after exercise as compared to the baseline values and at 60 min after (61.16%).

However, in the control group we observed a gradual decrease, with the highest cfDNA level prior to exercise. This was quite surprising. On the other hand, the reason for that may be associated with the specificity of maximal physical activity testing. Contrary to athletes, untrained people may not have developed the ability to use all of the body’s psychophysical resources to undertake maximal anaerobic effort, and thus potentially lower potential damage to muscle tissue. On the other hand, it was already observed [13] that psychosocial stress exposure as well as physical exercise lead to increased cfDNA release. That may be the reason why the control group had elevated cfDNA levels before their first maximal physical effort and it could have limited their engagement in the MAnE effort. After they finished MAnE, they relaxed, mental stress subsided and that could lead to diminished cfDNA plasma concentration. However, this is only a speculation and theoretical guess; because, to confirm it, the concentration of stress markers—such as cortisol, α amylase, testosterone, and others—would also need to be assessed. As a result of the above changes, 68.47% higher cfDNA concentrations were also noticed up to 5 min after MAnE in the athlete group compared to the control group.

Fatouros at al. [9] observed elevated cfDNA levels even 96 h after exercise among men who took part in a 12 week, low-, high- and very high intensity resistance training program. Those results are quite similar to our analyses, despite the fact that the time points are different. We measured plasma cfDNA at rest, up to 5, 30, and 60 min after two different maximal physical efforts to look for cfDNA changes associated with double repeated Wingate Anaerobic Test and VO2max Bruce testing in groups of athletes and physical active men. Our analysis also showed a significant increase of cfDNA concentration in the group of athletes up to 5 min after MAE and the concentration of cfDNA remained at the same state for longer period. Elevated cfDNA already lasted up to 3 h after severe injury in trauma patients [33], 2 or more days after exhaustive exercise [8], and even up to several days in patients suffering multiple organ dysfunction syndromes [34]. Therefore, analyses cfDNA in our participants over longer time intervals would probably lead to the same conclusions.

Furthermore, typical markers of skeletal muscle damage (C-reactive protein, creatine kinase, myoglobin) display delayed kinetics compared with the cfDNA peak response. All of that underlines the great potential of cfDNA as a biomarker for exercise load in both the aerobic and the anaerobic state.

Plasma cfDNA measurement is non-invasive and requires limited time and personnel. Further investigation of the origin of plasma cfDNA in the overtrained state is warranted.

In conclusion, this is the first report demonstrating that maximal aerobic and anaerobic exercise significantly increases plasma levels of cfDNA, possibly resulting from early stages of tissue inflammation reaction and as a predictor of further activity processes and the stimulating effect of physical activity on body metabolism. These observations imply that formation of cfDNA is associated with occurrences of physical activity but only in maximal physical activity. Plasma cfDNA measurement similar to exerkine analyses is additionally non-invasive and requires limited time and personnel to receive specific data. Further investigation of the origin of plasma cfDNA in the blood stream associated with muscle work should be undertaken.

## 5. Limitations

The source of elevated levels of cfDNA in the bloodstream still remains speculative, whether this is a rapid cfDNA accumulation due to active mechanisms or passive cell death events needs further analysis. Elevated cfDNA level should be associated with selected exerkine levels for wide spectrum immunological response analyses (increased plasma level of CRP, CK, interleukins). Furthermore, analyses in specific adapted sport populations should be performed to show sport dependent mechanism of adaptation.

## 6. Practical Application

Based on the obtained results, it could be suggested that for the maintenance of adaptation on the molecular level, cfDNA measurement could be performed. It could be useful information for sport practice, which indicates early stages of immunological response associated with very intensive physical activity.

## Figures and Tables

**Figure 1 genes-12-01238-f001:**
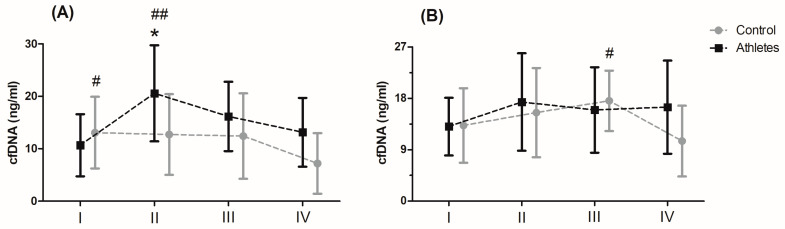
Concentration of cell-free DNA (cfDNA) after the maximal anaerobic (**A**) and aerobic effort (**B**) in training and control group (means and standard deviations). I, baseline; II, up to 5 min after the effort; III, 30 min after the effort; IV, 60 min after the effort. Significant difference at *p* < 0.05 vs: * II-Controls; # IV-Control; ## I, IV—Athletes.

**Figure 2 genes-12-01238-f002:**
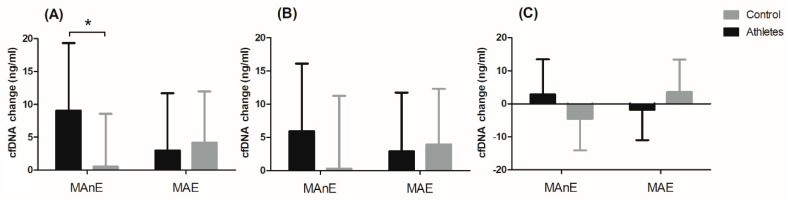
Changes in cell-free DNA (cfDNA) from baseline to u to 5 min after (**A**) and 30 min (**B**) and 60 min (**C**) after maximal anaerobic (MAnE) and aerobic effort (MAE) in the athletes and controls. * significant difference vs. control group at *p* < 0.01.

**Table 1 genes-12-01238-t001:** Physical characteristics of the participants (*n* = 50).

Variables	Unit	Control (*n* = 25)		Athletes (*n* = 25)		*p*
		Mean ± SD	CI	Mean ± SD	CI	
Height	cm	180.12 ± 6.46	177.81–184.30	178.60 ± 6.1	176.08–181.03	0.12
Body Mass	kg	79.40 ± 13.34	74.83–83.69	76.03 ± 6.78	73.66–81.95	0.26
Skeletal Muscle Mass	kg	39.45 ± 5.64	38.07–41.71	39.40 ± 3.62	37.59–41.75	0.97
Body Fat Mass	kg	10.37 ± 6.18	7.66–12.46	7.88 ± 2.75	6.91–10.08	0.07
Percent Body Fat	%	12.48 ± 5.87	9.81–14.56	10.25 ± 2.75	9.02–12.47	0.10
Body Mass Index	kg/m^2^	23.69 ± 3.44	22.30–25.08	23.95 ± 1.63	23.15–25.70	0.65

## Data Availability

The data presented in this study are available on request from the corresponding author. The data are not publicly available due to ethical reasons.

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
