# Peer review of "cfDNA Changes in Maximal Exercises as a Sport Adaptation Predictor"

_genes, 2021, doi:10.3390/genes12081238_

Round 1

Reviewer 1 Report

Manuscript titled “cfDNA changes in maximal exercises as a sport adaptation predictor” aims to deep the knowledge on the increase of cfDNA in response to different kind of exercises.

This represents an innovative and original research, and it could lay the basis for further researches and practical applications in sport sciences.

The manuscript reports some interesting and original data, however, there are some minor points should be modified:

Introduction:

line 44: since author mention “rising evidence”, please add proper references;

line 87: add references;

 Materials and methods:

Please add little paragraph describing methodology for detecting body composition (fat mass and skeletal muscle mass), or, if they are not relevant for the study, eliminate them from work. In my opinion, this information could be useful for asserting a homogeneity in body composition between the two groups, or at least for including all of them in normal weight group.

Line 166-167: are the time-points for blood collection standardized? Authors should explain why they choose these intervals.

Line 197: since in tables authors indicate p value, they should indicate p instead of alpha, if no further correction was done;

Results and discussion

In my opinion it could be important to test and discuss the variability within each group, to test a homogeneity or heterogeneity in the response, and to verify the presence of further factors involved in the response.

Reviewer 2 Report

This study reports the effects of an anaerobic and aerobic test on cell-free DNA in 25 control subjects and 25 footballers before and at several time points after exercise. The main data are presented in Figure 1 and show that the cfDNA appears to increase in athletes but not controls after anaerobic exercise and in no group after aerobic exercise.  The relatively large cohort and the measurement of cfDNA at different time points is a strength of this study. 

Major issues are:

  • Some statements overinterpret the data. For example, the abstract states "cfDNA may be a predictor for adaptation for anaerobic exercise". The authors have not conducted a training study and so there is no basis for this statement". Also, the statements in relation to overtraining e.g. in the abstract are "bold" as the authors have not attempted to investigate overtraining.  
  • The authors present a lot of data that are not really central to the cfDNA story e.g. in tables 1-4. These data could go in the supplementary data and key data could be integrated into sentences that briefly describe the subjects. Also in table 2 the variables should be spelled out, if at all possible because it is a pain to always go back to the legend to find out what e.g. PLT means.  
  • The cfDNA data are presented several times. First in Figure 1, then the changes in Figure 2 and the statistics in table 6. This could be condensed to one figure that ideally shows the data of each individual without being overcrowded
  • There are factual errors. For example, cancer is not an autoimmune disorder (line 58).

Overall, these are potentially interesting data. However, the manuscript should be shorter but more focused and data that are not essential to the story can be put into the supplementary data. Also, to study cfDNA as a damage marker it would be good to correlate cfDNA with creatine kinase, a clearly muscle-derived damage marker. 

Round 2

Reviewer 2 Report

Thanks to the authors for addressing the comments. The English is still limited and a native speaker should check the proofs. My final request is to change the title:  "cfDNA changes in maximal exercises as a sport adaptation predictor" The term "in maximal exercises" is not correct English and the authors have not validated cfDNA as a sport adaptation predictor as they have not conducted a training study. 

One idea for a title could be: "cfDNA increases during maximal but not submaximal exercise" Again, the authors should have a native speaker check their title.